# Label Noise in Adversarial Training: A Novel Perspective to Study Robust Overfitting

**Chengyu Dong**
University of California, San Diego
cdong@eng.ucsd.edu

**Liyuan Liu**
Microsoft Research
lucliu@microsoft.com

**Jingbo Shang**
University of California, San Diego
jshang@eng.ucsd.edu

## Abstract

We show that label noise exists in adversarial training. Such label noise is due to the mismatch between the true label distribution of adversarial examples and the label inherited from clean examples – the true label distribution is distorted by the adversarial perturbation, but is neglected by the common practice that inherits labels from clean examples. Recognizing label noise sheds insights on the prevalence of robust overfitting in adversarial training, and explains its intriguing dependence on perturbation radius and data quality. Also, our label noise perspective aligns well with our observations of the epoch-wise double descent in adversarial training. Guided by our analyses, we proposed a method to automatically calibrate the label to address the label noise and robust overfitting. Our method achieves consistent performance improvements across various models and datasets without introducing new hyper-parameters or additional tuning.

## 1 Introduction

Adversarial training (Goodfellow et al., 2015; Huang et al., 2015; Kurakin et al., 2017; Madry et al., 2018) is known as one of the most effective ways (Athalye et al., 2018; Uesato et al., 2018) to enhance the adversarial robustness of deep neural networks (Szegedy et al., 2014; Goodfellow et al., 2015). It augments training data with adversarial perturbations to prepare the model for adversarial attacks. Despite various efforts to generate more effective adversarial training examples (Ding et al., 2020; Zhang et al., 2020), the labels assigned to them attracts little attention. As the common practice, the assigned labels of adversarial training examples are simply inherited from their clean counterparts.

In this paper, we argue that the existing labeling practice of the adversarial training examples introduces label noise implicitly, since adversarial perturbation can distort the data semantics (Tsipras et al., 2019; Ilyas et al., 2019). For example, as illustrated in Figure 1, even with a slight distortion of the data semantics (e.g., more ambiguous), the label distribution of the adversarially perturbed data may not match the label distribution of the clean counterparts. Such distribution shift is neglected when assigning labels to adversarial examples, which are directly copied from the clean counterparts. We observe that distribution mismatch caused by adversarial perturbation along with improper labeling practice will cause *label noise in adversarial training*.

It is a mysterious and prominent phenomenon that the robust test error would start to increase after conducting adversarial training for a certain number of epochs (Rice et al., 2020), and our label noise perspective provides an adequate explanation for this phenomenon. Specifically, from a classic bias-variance view of model generalization, label noise that implicitly exists in adversarial training

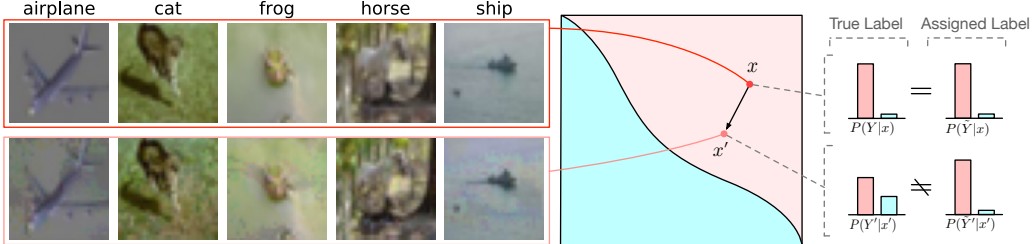

Figure 1: Illustration of the origin of label noise in adversarial training. The adversarial perturbation causes a mismatch between the true label distributions of clean inputs $x$ and their adversarial examples $x'$. Such a distribution mismatch is however neglected by the labels assigned to adversarial examples in the common practice of adversarial training, resulting in label noise implicitly.

can increase the model variance (Yang et al., 2020) and thus make the overfitting much more evident compared to standard training. Further analyses of label noise in adversarial training also explain the intriguing dependence of robust overfitting on the perturbation radius (Dong et al., 2021b) and data quality (Dong et al., 2021a) presented in the literature.

Providing the label noise in adversarial training, one can further expect the existence of double descent based on the modern generalization theory of deep neural networks. *Epoch-wise double descent* refers to the phenomenon that the test error will first decrease and then increase as predicted by the classic bias-variance trade-off, but it will decrease again as the training continues. Such phenomenon is only reported in standard training of deep neural networks, often requiring significant label noise in the training set (Nakkiran et al., 2020). As the label noise intrinsically exists in adversarial training, such epoch-wise double descent phenomenon also emerges when the training goes longer. Indeed, as shown in Figure 2, for a relatively large model such as WRN-28-5, on top of the existing robust overfitting phenomenon, the robust test error will eventually decrease again *after* $1,000$ *epochs*. Following Nakkiran et al. (2020), we further experiment different model sizes. One can find that a medium-sized model will follow a classic U-curve, which means only overfitting is observed; and the robust test error for a small model

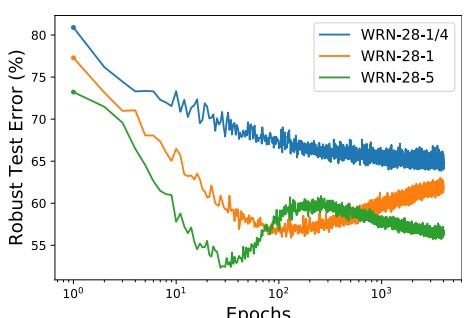

Figure 2: Robust overfitting can be viewed as an early part of the epoch-wise double descent. We employ PGD training (Madry et al., 2018) on CIFAR-10 (Krizhevsky, 2009) with Wide ResNet (WRN) (Zagoruyko & Komodakis, 2016) and a fixed learning rate. WRN-28-$k$ refers to WRN with depth $28$ and widen factor $k$.

will monotonically decrease. These are well aligned with the observations in standard training regime. This again consolidates our understanding of label noise in adversarial training.

In light of our analyses, we design a theoretically-grounded method to mitigate the label noise in adversarial training automatically. The key idea is to resort to an alternative labeling of the adversarial examples. We show that the predictive label distribution of an adversarially trained probabilistic classifier can approximate the true label distribution with high probability. Thus it can be utilized as a better labeling of the adversarial examples and provably reduce the label noise. We also show that with proper temperature scaling and interpolation, such predictive label distribution can further reduce the label noise. This echoes the recent empirical practice of incorporating knowledge distillation (Hinton et al., 2015) into adversarial training (Chen et al., 2021). While previous works heuristically select fixed scaling and interpolation parameters for knowledge distillation, we show that it is possible to fully unleash the potential of knowledge distillation by automatically determining the set of parameters that maximally reduces the label noise, with a strategy similar to confidence calibration (Guo et al., 2017). Such strategy can further mitigate robust overfitting to a minimal amount without additional human tuning effort. Extensive experiments on different datasets, training methods, neural architectures and robustness evaluation metrics verify the effectiveness of our method.

In summary, our findings and contributions are: 1) we show that the labeling of adversarial examples in adversarial training practice introduces label noise implicitly; 2) we show that robust overfitting

can be adequately explained by such label noise, and it is the early part of an epoch-wise double descent; 3) 2e show an alternative labeling of the adversarial examples can be established to provably reduce the label noise and mitigate the robust overfitting.

## 2 Related Work

**Robust overfitting and double descent in adversarial training.** Double descent refers to the phenomenon that overfitting by increasing model complexity will eventually improve test set performance (Neyshabur et al., 2017; Belkin et al., 2019). This appears to conflict with the robust overfitting phenomenon in adversarial training, where increasing model complexity by training longer will impair test set performance constantly after a certain point during training. It is thus believed in the literature that robust overfitting and epoch-wise double descent are separate phenomena (Rice et al., 2020). In this work we show this is not the complete picture by conducting adversarial training for exponentially more epochs than the typical practice.

A recent work also considers a different notion of double descent that is defined with respect to the perturbation size (Yu et al., 2021). Such double descent might be more related to the robustness-accuracy trade-off problem (Papernot et al., 2016; Su et al., 2018; Tsipras et al., 2019; Zhang et al., 2019), rather than the classic understanding of double descent based on model complexity.

**Mitigate robust overfitting.** Robust overfitting hinders the practical deployment of adversarial training methods as the final performance is often sub-optimal. Various regularization methods including classic approaches such as $\ell_1$ and $\ell_2$ regularization and modern approaches such as cutout (Devries & Taylor, 2017) and mixup (Zhang et al., 2018) have been attempted to tackle robust overfitting, whereas they are shown to perform no better than simply early stopping the training on a validation set (Rice et al., 2020). However, early stopping raises additional concern as the best checkpoint of the robust test accuracy and that of the standard accuracy often do not coincide (Chen et al., 2021), thus inevitably sacrificing the performance on either criterion. Various regularization methods specifically designed for adversarial training are thus proposed to outperform early stopping, including regularization the flatness of the weight loss landscape (Wu et al., 2020; Stutz et al., 2021), introducing low-curvature activation functions (Singla et al., 2021), data-driven augmentations that adds high-quality additional data into the training (Rebuffi et al., 2021) and adopting stochastic weight averaging (Izmailov et al., 2018) and knowledge distillation (Hinton et al., 2015) (Chen et al., 2021). These methods are likely to suppress the label noise in adversarial training, with the self-distillation framework (i.e. the teacher shares the same architecture as the student model) introduced by (Chen et al., 2021) as a particular example since introducing teacher's outputs as supervision is almost equivalent to the alternative labeling inspired by our understanding of the origin of label noise in adversarial training.

## 3 Preliminaries

**A statistic model of label noise (Frénay & Verleysen, 2014).** Let $\mathcal{X} \subset \mathbb{R}^d$ define the input space equipped with a norm $\| \cdot \| : \mathcal{X} \to \mathbb{R}^+$ and $\mathcal{Y} = [K] := \{1, 2, \dots, K\}$ define the label space. We introduce four random variables to describe noisy labeling process. Let $X \in \mathcal{X}$ denote the input, $Y \in \mathcal{Y}$ denote the true label of the input, $\tilde{Y} \in \mathcal{Y}$ denote the assigned label of an input provided by an annotator, and finally $E$ denote the occurrence of a label error by this annotator. $E$ is a binary random variable with value 1 indicating that the assigned label is different from the true label for a given input, i.e., $E = \mathbf{1}(\tilde{Y} \neq Y)$. We study the case where the label error depends on both the input $X$ and the true label $Y$. For a classification problem, a training set consists of a set of examples that are sampled as $\mathcal{D} = \{(x_i, \tilde{y}_i)\}_{i \in [N]}$.

**Definition 3.1** (Label noise). *We define label noise $p_e$ in a training set $\mathcal{D}$ as the empirical measure of the label error, namely $p_e(\mathcal{D}) = 1/N \sum_{i \in [N]} \mathbf{1}(\tilde{y}_i \neq y_i)$.*

**Assumption 3.1.** *We assume the annotation of a clean dataset involves no label error, namely $P(E = 1 | Y = y, x) = 0$. This directly implies $P(\tilde{Y} | x) = P(Y | x)$ (see proof in the Appendix).*

**Definition 3.2** (Data quality). *Given a training set $\mathcal{D}$, we define its data quality as $q(\mathcal{D}) = \mathbb{E}_{(x,y) \in \mathcal{D}} P(Y = y | x)$*

**Adversarially augmented training set.** Let $f : \mathcal{X} \rightarrow \mathcal{Y}$ be a probabilistic classifier and $f(\cdot)_j$ be its predictive probability at class $j$. The adversarial example of $x$ generated by $f$ is obtained by solving the maximization problem $x' = \arg\max_{z \in \mathcal{B}_\varepsilon(x)} \ell(f(z), y)$. Here $\ell$ can be a typical loss function such as cross-entropy. And $\mathcal{B}_\varepsilon(x)$ denotes the norm ball centered at $x$ with radius $\varepsilon$, i.e., $\mathcal{B}_\varepsilon(x) = \{z \in \mathcal{X} : \|z - x\| \leq \varepsilon\}$.

Following previous notations, we denote $Y'$ as a random variable representing the true label of $x'$ and $\tilde{Y}'$ as a random variable representing the assigned label of $x'$. We refer $\mathcal{D}' = \{(x', \tilde{y}')\}$ as the adversarially augmented training set.

**Adversarial training.** Adversarial training can be viewed as a data augmentation technique that trains the parametric classifier $f_\theta$ on the adversarially augmented training set (Tsipras et al., 2019), namely

$$\theta^* = \arg\min_\theta \frac{1}{|\mathcal{D}'|} \sum_{(x', \tilde{y}') \in D'} \ell(f_\theta(x'), \tilde{y}'). \tag{1}$$

# 4   Label noise implicitly exists in adversarial training

In this section, we demonstrate the implicit existence of label noise in the adversarially augmented training set. We first consider a simple case where the adversarial perturbation is generated based on an ideal classifier that predicts the true label distribution. Under such a case we prove that the label noise in the adversarially augmented training set is lower-bounded. We then show that in realistic cases an adversarially trained classifier can approximate the true label distribution with high probability. Therefore, additional error terms will be required to lower bound the label noise. All proofs for the remainder of this paper are provided in the appendix.

## 4.1   When adversarial perturbation is generated by the true probabilistic classifier

We first consider an ideal case where the adversarial perturbation is generated by the true probabilistic classifier $f(x) := P(Y|x)$, namely the classifier producing the true label distribution on any input $x$.

**The true label distribution is distorted by adversarial perturbation.** We quantify the mismatch between two probability distributions using the *total variation (TV) distance*.

**Definition 4.1** (TV distance). *Let $\mathcal{A}$ be a collection of the subsets of the label sample space $\mathcal{Y}$. The TV distance between two probability distributions $P(Y)$ and $P(Y')$ can be defined as $\|P(Y) - P(Y')\|_{TV} = \sup_{J \in \mathcal{A}} |P(Y \in J) - P(Y' \in J)|$.*

We now show that adversarial perturbation generated by the true probabilistic classifier can induce a mismatch between the true label distributions of clean inputs and their adversarial examples. For simplicity we consider adversarial perturbation based on FGSM and cross-entropy loss, namely $x' = x - \varepsilon\|\nabla f(x)_y\|^{-1}\nabla f(x)_y$. The distribution mismatch induced by such adversarial perturbation can be lower bounded.

**Lemma 4.1.** *Assume $f(x)_y$ is L-locally Lipschitz around $x$ with bounded Hessian. Let $\sigma_m = \inf_{z \in \mathcal{B}_\varepsilon(x)} \sigma_{\min}(\nabla^2 f(z)_y) > 0$ and $\sigma_M = \sup_{z \in \mathcal{B}_\varepsilon(x)} \sigma_{\max}(\nabla^2 f(z)_y) > 0$. Here $\sigma_{\min}$ and $\sigma_{\max}$ denote the minimum and maximum eigenvalues of the Hessian, respectively. We then have*

$$\|P(Y|x) - P(Y'|x')\|_{TV} \geq \frac{\varepsilon}{2}(1 - f(x)_y)\frac{\sigma_m}{L} - \frac{\varepsilon^2}{4}\sigma_M, \tag{2}$$

One can find that the right-hand side is positive as long as the upper bound of the Hessian norm is not too large, which is reasonable as previous works have shown that small hessian norm is critical to both standard (Keskar et al., 2017) and robust generalization (Moosavi-Dezfooli et al., 2019).

**Assigned label distribution is unchanged.** Despite the fact that the true label distribution is distorted by adversarial perturbation, we note that the assigned label distribution of adversarial examples is still the same as their clean counterparts.

**Remark 4.1.** *In adversarial training, it is the common practice that directly copies the label of a clean input to its adversarial counterpart, namely $\tilde{y}' = \tilde{y}$ and $P(\tilde{Y}'|x') = P(\tilde{Y}|x)$.*

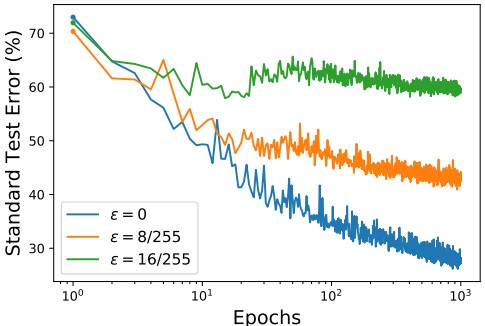
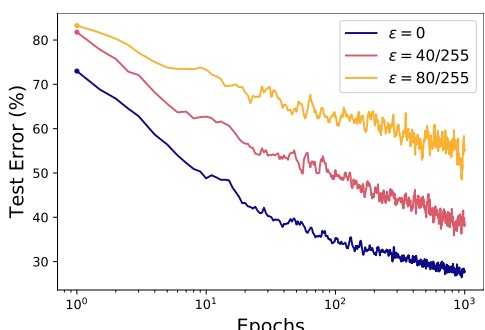

Figure 3: Standard training on a fixed adversarially augmented training set (e.g. $\varepsilon = 16/255$) can also produce prominent overfitting. In contrast, on the original training set without adversarial perturbation applied ($\varepsilon = 0$), no overfitting is observed.

Figure 4: Standard training on a training set augmented by Gaussian noise will not produce overfitting. Here we select extremely large perturbation radius (e.g. $\varepsilon = 80/255$) to reduce the test error to be comparable to the adversarially augmented case.

**Distribution mismatch indicates label noise.** We show that a mismatch between the true label distribution and the assigned label distribution in a training set will always indicate the existence of label noise.

**Lemma 4.2.** *Given a training set $\mathcal{D} = \{(x_i, \tilde{y}_i)\}_{i \in [N]}$, the label noise is lower-bounded by the mismatch between the true label distribution and the assigned label distribution. Specifically, with probability $1 - \delta$, we have*

$$p_e(\mathcal{D}) \geq \mathbb{E}_x \|P(\tilde{Y}|x) - P(Y|x)\|_{TV} - \sqrt{\frac{1}{2N} \log \frac{2}{\delta}} \tag{3}$$

**Label noise implicitly exists in adversarial training.** In the adversarially augmented training set $\mathcal{D}'$, such distribution mismatch exists exactly. By Remark 4.1 we have $P(\tilde{Y}'|x') = P(\tilde{Y}|x)$ and by property of the clean dataset (Assumption 3.1) we have $P(\tilde{Y}|x) = P(Y|x)$, which together means $P(\tilde{Y}'|x') = P(Y|x)$. However, Lemma 4.1 shows that $P(Y'|x') \neq P(Y|x)$, which implies that $P(\tilde{Y}'|x') \neq P(Y'|x')$. This indicates that label noise exists in the adversarially augmented training set. We now have the following theorem, which is our main result.

**Theorem 4.3.** *Assume $f(x)_y$ is L-locally Lipschitz around $x$ with Hessian bounded below. Instantiate the same notations as in Lemma 4.1. With probability $1 - \delta$, we have*

$$p_e(\mathcal{D}') \geq \frac{\varepsilon}{2}(1 - q(\mathcal{D}))\frac{\sigma_m}{L} - \frac{\varepsilon^2}{4}\sigma_M - \sqrt{\frac{1}{2N} \log \frac{2}{\delta}} \tag{4}$$

The above results suggest that as long as a training set is augmented by adversarial perturbation, but with assigned labels unchanged, label noise emerges. We demonstrate this by showing that standard training on a fixed adversarially augmented training set can also produce overfitting. Specifically, for each example in a clean training set we apply adversarial perturbation generated by a adversarially trained classifier. We then fix such an augmented training set and conduct standard training on it. We experiment on CIFAR-10 with WRN-28-5. A training subset of size 5k is randomly sampled to speed up the training. More details about the experiment settings can be found in the appendix. Figure 3 shows that prominent overfitting (as well as epoch-wise double descent) can be observed when the perturbation radius is relatively large.

On the other hand, if a training set is augmented by perturbation that will not distort the true label distribution, there will not be label noise. We demonstrate this by showing that standard training on a training set augmented with Gaussian noise will not induce overfitting. As shown in Figure 4, even with a extremely large radius of Gaussian perturbation, no overfitting is observed. This also demonstrates that input perturbation not necessarily leads to overfitting.

**Intuitive interpretation of label noise in adversarial training.** We introduce a simple example to help understand the emergence of label noise in adversarial training.

**Example 4.1** (Label noise due to a symmetric distribution shift). *Let $\mathcal{D} = \{(x_i, y_i)\}_{i \in [N]}$ be a clean labeled training subset where all inputs $x_i = x$ are identical and have a one-hot true label distribution, i.e., $P(Y|x) = \mathbf{1}_y$.*

*We now construct an adversarially augmented training subset $\mathcal{D}' = \{(x_i', \tilde{y}_i')\}_{i \in [N]}$, where $\tilde{y}' = y$ and $x'$ is generated based on adversarial perturbation that distorts the true label distribution symmetrically. Specifically,*

$$P(Y' = j'|x') = \begin{cases} 1 - \eta, & \text{if } j = y, \\ \eta/(K - 1), & \text{otherwise.} \end{cases}$$

*Then by Lemma 4.2 we have $p_e(\mathcal{D}') \gtrsim \eta$.*

One can find that there is indeed $\eta$ faction of noisy labels in $D'$. This is because if we sample the labels of $x'$ based on its true label distribution, we expect $1 - \eta$ faction of $x'$ are labeled as $y$, while $\eta$ fraction of $x'$ are labeled to be other classes. However, in $D'$, all $x'$ are assigned with label $y$, which means $\eta$ fraction of $x'$ are labeled incorrectly. In realistic datasets we can consider inputs with similar features for such reasoning.

The above example also shows that label noise in adversarial training may be stronger than one's impression. Even a slight distortion of the true label distribution, e.g. $\eta = 0.1$, will be equivalent to at least 10% noisy label in the training set. This is because the true label distribution of every training input is distorted, resulting in significant noise in the population.

**Dependence of label noise in adversarial training.** Theorem 4.3 shows that the label noise in adversarial training is proportional to (1) the perturbation radius (2) the data quality. Considering label noise can be an important source of variance in the generalization of deep neural networks (Nakkiran et al., 2020; Yang et al., 2020), such dependence of label noise explains the intriguing observations in the literature that robust overfitting (or epoch-wise double descent) in adversarial training will vanish with small perturbation radii (Dong et al., 2021b) or high-quality data (Dong et al., 2021a). We conduct more controlled experiments to verify this correlation empirically, as shown in Figure 5.

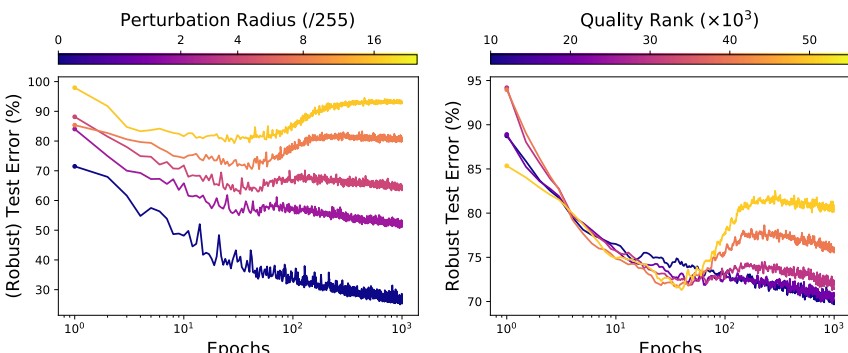

Figure 5: (Left) Dependence of robust overfitting on the perturbation radius. A training subset of size 5k is randomly sampled to speed up the training. $\varepsilon = 0/255$ indicates the standard training where no double descent occurs. (Right) Dependence of robust overfitting on the data quality with a fixed perturbation radius ($\varepsilon = 8/255$). To construct a training subset with high data quality, we first calculate the predictive probability based on an ensemble of multiple models. We then rank all training examples based on the predictive probability and select the top-k ones. The curves are smoothed by a window of 5 epochs to reduce overlapping. Here we conduct PGD training on CIFAR-10 with WRN-28-5. More experiment details can be found in the Appendix.

### 4.2 Adversarial perturbation generated by a realistic classifier

We now consider a realistic case where the adversarial perturbation is generated by a probabilistic classifier $f_\theta$.

**Approximation of the true label distribution.** We show that after sufficient adversarial training, the predictive label distribution of $f_\theta$ can approximate the true label distribution with high probability.

**Lemma 4.4.** *Denote $\mathcal{S} = \{x : (x, y) \in \mathcal{D}\}$ as the collection of all training inputs. Let $\rho \geq 1$ and $\mathcal{C}$ be an $\rho\varepsilon$-external covering of $\mathcal{S}$ with covering number $N_{\rho\varepsilon}$. Let $f_\theta$ be a probabilistic classifier*

*that minimizes the adversarial empirical risk ([1]). Assume $f_\theta$ is $L_\theta$-locally Lipschitz continuous in a norm ball of radius $\rho\varepsilon$ around $x \in \mathcal{C}$. Let $\kappa \geq 1$ and $\hat{\mathcal{S}}$ be a subset of $\mathcal{S}$ with cardinality at least $(1-1/\kappa+1/(\kappa N_{\rho\varepsilon}))N$. Let $\mathcal{N}_\varepsilon(\hat{\mathcal{S}})$ denote the neighborhood of the set $\hat{\mathcal{S}}$, i.e. $\mathcal{N}_\varepsilon(\hat{\mathcal{S}}) = \bigcup_{x\in\hat{\mathcal{S}}} \mathcal{B}_\varepsilon(x)$. Then for any $x \in \mathcal{N}_\varepsilon(\hat{\mathcal{S}})$, with probability at least $1-\delta$,*

$$\|f_\theta(x) - P(Y|x)\|_{TV} \leq \sqrt{\frac{\kappa N_{\rho\varepsilon}K}{2N}\log\frac{2}{\delta}} + \left(\left(\frac{3}{2}-\frac{1}{K}\right)L_\theta + L\right)\rho\varepsilon, \quad (5)$$

**Label noise in adversarial training with a realistic classifier.** Adversarial perturbation generated by a realistic classifier $f_\theta$ will distort its predictive label distribution by gradient ascent. Subsequently, the true label distribution will also be distorted with high probability since the predictive label distribution of a realistic classifier $f_\theta$ can approximate the true label distribution. Specifically, by the triangle inequality we have

$$\|P(Y|x) - P(Y'|x')\|_{\text{TV}} \geq \|f_\theta(x) - f_\theta(x')\|_{\text{TV}} - (\|f_\theta(x) - P(Y|x)\|_{\text{TV}} + \|f_\theta(x') - P(Y'|x')\|_{\text{TV}}),$$

$$(6)$$

where the last two terms are the approximation error of true label distribution on both clean and adversarial examples, which are guaranteed to be small. To conclude, we have the following result.

**Theorem 4.5.** *Instantiate the notations of Lemma [4.4]. For any $x \in \mathcal{N}_\varepsilon(\hat{\mathcal{S}})$, with probability at least $1 - 3\delta$, we have*

$$p_e(\mathcal{D}') \geq \varepsilon\left[(1 - \mathbb{E}_x f_\theta(x)_y)\frac{\sigma_m}{2L_\theta} - 2\rho\left(\left(\frac{3}{2}-\frac{1}{K}\right)L_\theta + L\right)\right] - \varepsilon^2\frac{\sigma_M}{4} - \xi\sqrt{\frac{1}{2N}\log\frac{2}{\delta}}, \quad (7)$$

*where $\xi = 1 + \sqrt{4\kappa N_{\rho\varepsilon}K}$.*

# 5 Mitigate Label Noise in Adversarial Training

Since the label noise is incurred by the mismatch between the true label distribution and assigned label distribution of adversarial examples in the training set, we wish to find an alternative label (distribution) for the adversarial example to reduce such distribution mismatch. We've already shown that the predictive label distribution of a classifier trained by conventional adversarial training, which we denote as *model probability* in the following discussion, can in fact approximate the true label distribution. Here we show that it is possible to further improve the predictive label distribution and reduce the label noise by calibration.

## 5.1 Rectify model probability to reduce distribution mismatch

We show that it is possible to reduce the distribution mismatch by *temperature scaling* (Hinton et al., 2015; Guo et al., 2017) enabled in the softmax function.

**Theorem 5.1** (Temperature scaling can reduce the distribution mismatch). *Let $f_\theta(x;T)$ denote the predictive probability of a probabilistic classifier scaled by temperature $T$, namely $f_\theta(x;T)_j = \exp(z_j/T)/(\sum_j \exp(z_j/T))$, where $z$ is the logits of the classifier from $x$. Let $x'$ be an adversarial example correctly classified by a classifier $f_\theta$, i.e. $\arg\max_j f_\theta(x')_j = y'$, then there exists $T$, such that*

$$\|f_\theta(x';T) - P(Y'|x')\|_{TV} \leq \|f_\theta(x') - P(Y'|x')\|_{TV}.$$

Another way to further reduce the distribution mismatch is to interpolate between the model probability and the one-hot assigned label. We show that the interpolation works specifically for incorrectly classified examples and thus can be viewed as a complement to temperature scaling.

**Theorem 5.2** (Interpolation can further reduce the distribution mismatch). *Let $x'$ be an adversarial example incorrectly classified by a classifier $f_\theta$, i.e. $\arg\max_j f_\theta(x';T)_j \neq y'$. Assume $\max_j P(Y' = j|x') \geq 1/2$, then there exists an interpolation ratio $\lambda$, such that*

$$\|f_\theta(x';T,\lambda) - P(Y'|x')\|_{TV} \leq \|f_\theta(x';T) - P(Y'|x')\|_{TV},$$

*where $f_\theta(x';T,\lambda) = \lambda \cdot f_\theta(x';T) + (1-\lambda) \cdot P(\tilde{Y}'|x')$.*

As a summarization, to reduce the distribution mismatch, we propose to use $f_\theta(x'; T, \lambda)$ as the assigned label of the adversarial example in adversarial training, which we refer as the *rectified model probability*.

In Appendix E, we show that the optimal hyper-parameters (i.e. $T$ and $\lambda$) of almost all training examples concentrate on the same set of values by studying on a synthetic dataset with known true label distribution. Therefore it is possible to find an universal set of hyper-parameters that reduce the distribution mismatch for all adversarial examples.

## 5.2 Determine the optimal temperature and interpolation ratio

The set of temperature and interpolation ratio in the rectified model probability that maximally reduces the distribution mismatch is not straightforward to find as the true label distribution of the adversarial example is unknown in reality. Fortunately, given a sufficiently large validation dataset as a whole, it is possible to measure the overall distribution mismatch in a frequentist's view without knowing the true label distribution of every single example. A popular metric adopted here is the negative log-likelihood (NLL) loss, which is known as a proper scoring rule (Gneiting & Raftery, 2007) and is also employed in the confidence calibration of deep networks (Guo et al., 2017). By Gibbs's inequality it is easy to show that the NLL loss will only be minimized when the assigned label distribution matches the true label distribution (Hastie et al., 2001), namely

$$-\mathbb{E}_{(x',y')\in\mathcal{D}'_{\text{val}}} \log f_\theta(x'; T, \lambda)_{y'} \geq -\mathbb{E}_{P(Y)} P(Y'|x') \log P(Y'|x'). \tag{8}$$

Therefore, we propose to find the optimal $T$ and $\lambda$ as

$$T, \lambda = \underset{T,\lambda}{\arg\min} -\mathbb{E}_{(x',y')\in\mathcal{D}'_{\text{val}}} \log f_\theta(x'; T, \lambda)_{y'}. \tag{9}$$

## 5.3 Rectified model probability mitigates robust overfitting

We now work on a realistic dataset (CIFAR-10) to demonstrate the rectified model probability can effectively mitigate the robust overfitting, or equivalently the epoch-wise double descent in adversarial training. The outer minimization of adversarial training (Equation (1)) now becomes

$$\theta^* = \underset{\theta}{\arg\min} \mathbb{E}_{\mathcal{D}'} \ell\left(f_\theta(x'), f_{\hat\theta}(x'; T, \lambda)_{y'}\right), \tag{10}$$

where $\hat\theta$ denotes the parameters of a classifier adversarially trained beforehand. The details of the experimental setting are available in the Appendix.

As shown in Figure 6, adversarial training on rectified model probability can mitigate the robust overfitting when the temperature $T$ and interpolation ratio $\lambda$ are optimal. Such optimal hyperparameters perfectly aligns with the ones automatically determined by Equation (9).

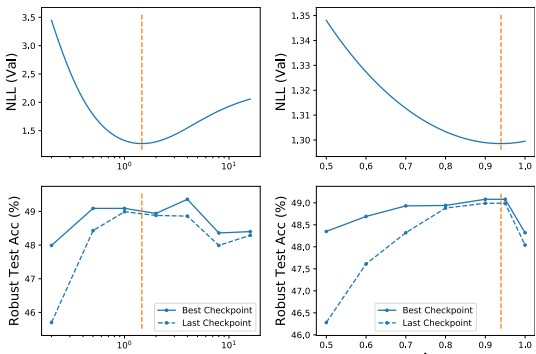

Figure 6: (Upper) NLL loss obtained on the validation set for different $T$ and $\lambda$. (Bottom) Robust test accuracy at the best and last checkpoint by adversarial training with the rectified model probability with different $T$ and $\lambda$. $\lambda = 0.8$ for grid search on $T$ (Left) and $T = 2$ for grid search on $\lambda$ (Right). Orange dashed lines indicate $T$ and $\lambda$ determined by Equation (9).

## 6 Experiments

**Experiment setup.** We conduct experiments on three datasets including CIFAR-10, CIFAR-100 (Krizhevsky, 2009) and Tiny-ImageNet (Le & Yang, 2015). We conduct PGD training on pre-activation ResNet-18 (He et al., 2016) with 10 iterations and perturbation radius $8/255$ by default. We evaluate robustness against $\ell_\infty$ norm-bounded adversarial attack with perturbation radius $8/255$, and employ AutoAttack (Croce & Hein, 2020) for reliable evaluation. Appendix D.2 includes results on additional model architectures (e.g., VGG (Simonyan & Zisserman, 2015), WRN), adversarial training

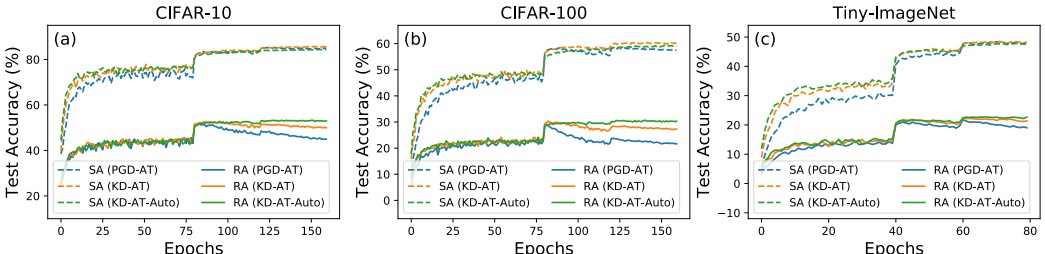

Figure 7: Our method can effectively mitigate robust overfitting for different datasets.

Table 1: Performance of our method on different datasets. $^*$ denotes the hyper-parameters automatically determined by our method.

| Dataset | Setting | $T$ | $\lambda$ | Robust Acc. (%) | | | Standard Acc. (%) | | |
|---|---|---|---|---|---|---|---|---|---|
| | | | | Best | Last | Diff. | Best | Last | Diff. |
| CIFAR-10 | AT | - | - | 47.35 | 41.42 | 5.93 | 82.67 | 84.91 | -2.24 |
| | KD-AT | 2 | 0.5 | 48.76 | 46.33 | 2.43 | 82.89 | **85.49** | -2.60 |
| | KD-AT-Auto | 1.47$^*$ | 0.8$^*$ | **49.05** | **48.80** | **0.25** | 84.26 | 84.47 | **-0.21** |
| CIFAR-100 | AT | - | - | 24.79 | 19.75 | 5.04 | 57.33 | 57.42 | -0.09 |
| | KD-AT | 2 | 0.5 | 25.77 | 23.58 | 2.19 | 57.24 | **60.04** | -2.80 |
| | KD-AT-Auto | 1.53$^*$ | 0.83$^*$ | **26.36** | **26.24** | **0.12** | 58.80 | 59.05 | **-0.25** |
| Tiny-ImageNet | AT | - | - | 17.20 | 15.40 | 1.80 | 47.72 | 47.62 | 0.10 |
| | KD-AT | 2 | 0.5 | 17.86 | 17.18 | 0.68 | **47.73** | **48.28** | -0.55 |
| | KD-AT-Auto | 1.23$^*$ | 0.85$^*$ | **18.29** | **18.39** | **-0.10** | 47.46 | 47.56 | **-0.10** |

methods (e.g., TRADES (Zhang et al., 2019), FGSM (Goodfellow et al., 2015)), and evaluation metrics (e.g., PGD-1000 (PGD attack with 1000 iterations), Square Attack (Andriushchenko et al., 2020), RayS (Chen & Gu, 2020)). More setup details can be found in Appendix G.

**Results & Discussions.**     Our method is essentially the baseline adversarial training with a robust-trained self-teacher, equipped with an algorithm automatically deciding the optimal hyper-parameters, which we now denote as KD-AT-Auto. We compare KD-AT-Auto with two baselines: regular adversarial training (AT), and adversarial training combined with self-distillation (KD-AT) with fixed temperature $T = 2$ and interpolation ratio $\lambda = 0.5$ as suggested by Chen et al. (2021).

As shown in Figure 7, our method can effectively mitigate robust overfitting for all datasets, with both standard accuracy (SA) and robust accuracy (RA) constantly increasing throughout training. In Table 1, we measure the difference between the RA at the best checkpoint (Best) and at the last checkpoint (Last) to clearly show the overfitting gap. Our method can reduce the overfitting gap to less than 0.5% for all datasets. One may note that self-distillation with fixed hyper-parameters is in fact inferior in terms of reducing robust overfitting, while its effectiveness can be significantly improved with the optimal hyper-parameters automatically determined by our method, which further verifies our understanding of robust overfitting. Compared with self-distillation with fixed hyper-parameters, our method can also boost both RA and SA at the best checkpoint for all datasets.

Our method can further be combined with orthogonal techniques such as Stochastic Weight Averaging (SWA) (Izmailov et al., 2018) and additional standard teachers as mentioned in previous work (Chen et al., 2021) to achieve better performance. More results and discussion can be found in Appendix D.3.

## 7   Conclusion and Discussions

In this paper, we show that label noise exists implicitly in adversarial training due to the mismatch between the true label distribution and the assigned label distribution of adversarial examples. Such label noise can explain the dominant overfitting phenomenon. Based on a label noise perspective, we also extend the understanding of robust overfitting and show that it is the early part of an epoch-wise double descent in adversarial training. Finally, we propose an alternative labeling of adversarial

examples by rectifying model probability, which can effectively mitigate robust overfitting without any manual hyper-parameter tuning.

The label noise implicitly exists in adversarial training may have other important effects on adversarially robust learning. This can potentially consolidate the theoretical grounding of robust learning. For instance, since label noise induces model variance, from a model-wise view, one may need to increase model capacity to reduce the variance. This may partially explain why robust generalization requires significantly larger model than standard generalization.

## Acknowledgments

We sincerely thank Like Hui for the discussions on the theory of double descent and confidence calibration. We thank anonymous reviewers for their valuable comments and suggestions. We also thank the audience at Neurips 2022 main conference for their thoughtful questions and feedback. Our work is sponsored in part by NSF Convergence Accelerator under award OIA-2040727, NIH Bridge2AI Center Program under award 1U54HG012510-01, as well as generous gifts from Google, Adobe, and Teradata. Any opinions, findings, and conclusions or recommendations expressed herein are those of the authors and should not be interpreted as necessarily representing the views, either expressed or implied, of the U.S. Government. The U.S. Government is authorized to reproduce and distribute reprints for government purposes not withstanding any copyright annotation hereon.

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
